

SciPost Phys. Lect. Notes 14 (2020)

# Notes on 8 Majorana fermions

**David Tong**[⋆] **and Carl Turner**[†]

Department of Applied Mathematics and Theoretical Physics,
University of Cambridge, Cambridge, CB3 0WA, UK

⋆ d.tong@damtp.cam.ac.uk, † c.p.turner@damtp.cam.ac.uk

## Abstract

Eight Majorana fermions in $d = 1 + 1$ dimensions enjoy a triality that permutes the representation of the $SO(8)$ global symmetry in which the fermions transform. This triality plays an important role in the quantization of the superstring, and in the analysis of interacting topological insulators and the associated phenomenon of symmetric mass generation. The purpose of these notes is to provide an introduction to the triality and its applications, with careful attention paid to various $\mathbf{Z}_2$ global and gauge symmetries and their coupling to background spin structures.

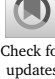

# 1 Introduction

A famous triality of $d = 1 + 1$ dimensional quantum field theory roughly states that eight Majorana fermions are the same as eight Majorana fermions which, in turn, are the same as eight Majorana fermions.

The essence of the triality is that the fermions in each theory transform in different representations of the $SO(8)$ global symmetry group. (Strictly speaking, this group is either Spin(8) or $SO(8)/\mathbf{Z}_2$ as we describe below.) If the fermions in the first theory transform in the vector representation $\mathbf{8_v}$, the fermions in the other two theories transform in the $\mathbf{8_s}$ and $\mathbf{8_c}$ spinor representations respectively. This means that if the fermions are coupled to a background $SO(8)$ gauge field, the classical theories look very different; the claim of triality is that, nonetheless, the quantum partition functions are the same.

This triality was first discovered by Shankar in the context of the Gross-Neveu model [1]. Subsequently, the triality played an important role in the quantization of the superstring [2], in particular in demonstrating the equivalence of the Green-Schwarz and Ramond-Neveu-Schwarz formulations on a torus [3]. The triality also underlies the work of Fidkowski and Kitaev [4], and subsequent developments [5–7] on interacting topological insulators.

Importantly, the triality is not an equivalence between free fermions. Instead, some of the theories must be coupled to one or more $\mathbf{Z}_2$ gauge fields. In the context of the superstring, this is what results in the need to sum over different spin structures and the associated GSO projection [8]. In the context of topological insulators, some aspects of these $\mathbf{Z}_2$ gauge fields were discussed in [9] and they play an important role in matching the phases of the theories across the duality.

Our purpose here to provide a self-contained, pedagogical review of the triality while, at the same time, paying attention to a number of subtleties that are usually swept under the rug. Most of these subtleties involve $\mathbf{Z}_2$ symmetries, both gauge and global, and the way in which these couple to chiral fermions and background spin structures.

**Free Fermions**

The need to consider such subtleties becomes clear if we review a few well known facts about free fermions. Take eight Majorana fermions, $\chi_i$, on the Lorentzian manifold $\mathbf{S}^1 \times \mathbf{R}$, with $\mathbf{S}^1$ the spatial circle. The action is simply

$$S_{\text{free}} = \int d^2x \ \sum_{i=1}^{8} i\bar{\chi}_i \slashed{\partial}_\rho \chi_i.$$

Here the subscript $\rho$ denotes the spin structure which, in the present case, simply tells us whether the fermions are periodic or anti-periodic around the circle.

The global symmetry group of the theory would naively appear to be $O(8)_L \times O(8)_R$, with the left- and right-moving fermions each transforming in the $\mathbf{8_v}$ representation of the appropriate subgroup. However, even in this simple theory things are not so straightforward.

In the case of anti-periodic boundary conditions (known as Neveu-Schwarz or NS in string theory), the fermion has a unique ground state. In this case, the spectrum of states is built above this ground state by acting with fermionic creation operators, and the spectrum does indeed fall into representations of $O(8)_L \times O(8)_R$. This precludes the possibility of any states sitting in $\mathbf{8_s}$ or $\mathbf{8_c}$ representations, for $O(8)$ has no such representations.

In the case of periodic boundary conditions (known as Ramond or simply R in string theory), the situation is different. Now there are a collection of Majorana zero modes for both left- and right-moving fermions which we denote as $\chi_{i,L}$ and $\chi_{i,R}$. They obey the commutation

relations

$$\{\chi_{i,L}, \chi_{j,L}\} = \{\chi_{i,R}, \chi_{j,R}\} = \delta_{ij} \qquad \text{and} \qquad \{\chi_{i,L}, \chi_{j,R}\} = 0. \tag{1.1}$$

We see that both left-movers and right-movers form a Clifford algebra. This means that the ground states do not lie in a representation of $SO(8)$, but rather of Spin(8).

In each of the left- and right-moving sectors, the zero modes build a $2^4 = 16$ dimensional representation which decomposes into the $\mathbf{8_s} \oplus \mathbf{8_c}$ irreducible spinor representations, distinguished by fermion number $(-1)^F$. The ground states then sit in the representation

$$(\mathbf{8_s} \oplus \mathbf{8_c}) \otimes (\mathbf{8_s} \oplus \mathbf{8_c}), \tag{1.2}$$

where the two factors are associated with the left- and right-movers respectively. On top of these ground states, excitations are built by acting with fermion creation operators, each of which transforms in the $\mathbf{8_v}$.

The upshot of this simple analysis is that the spectrum, and even the symmetry, depend strongly on the spin structure which dictates the periodicity of the fermions. Any putative dual theory should reproduce this behaviour, and so it too must be strongly sensitive to the spin structure. It is clear that it is not sufficient to simply consider free fermions and dictate by fiat that they transform in, say the $\mathbf{8_s}$ or $\mathbf{8_c}$ of Spin(8). Instead, something more interesting must be going on. The main goal of this paper is to review in some detail how this works.

**The Plan of the Paper**

There are, it turns out, a number of different trialities, related by gauging various $\mathbf{Z}_2$ symmetries. We start in Section 2 by describing 8 Majorana fermions coupled to a single $\mathbf{Z}_2$ gauge field. The resulting theory has a $SO(8)/\mathbf{Z}_2$ global symmetry and exhibits self-triality, a fact which is easily proven using bosonization techniques.

A slightly more involved version of triality yields a triumvirate of theories, one of which is the free fermion described above. In this case, the other theories in the triality orbit consist of eight Majorana fermions coupled to a $\mathbf{Z}_2 \times \mathbf{Z}_2$ chiral gauge field. These gauge fields, in turn, couple to the background spin structure through the Arf invariant, a topological invariant which plays a crucial role in other 2d dualities [10–15]. We describe this in Section 3.

Finally, we review two applications of the trialities. In Section 4 we describe the fermionic sector of the Type II and Type 0 superstrings, while in Section 5 we review the way in which interactions reduce the $\mathbf{Z}$ classification of certain low-dimensional topological insulators down to $\mathbf{Z}_8$. We also include two appendices. The first describes properties of the Arf invariant. This topological invariant, which can also be viewed as the mod 2 index of the chiral Dirac operator, plays an important role in coupling gauge fields to the background spin structure. The second appendix gives more details on the spectrum and symmetries of the trialities.

## 2 The Self-Triality of $SO(8)/\mathbf{Z}_2$

We first describe a theory that exhibits self-triality. This is a theory with neither $SO(8)$ nor Spin(8) global symmetry, but rather $SO(8)/\mathbf{Z}_2$. Before we describe the theory, we first review some simple group theory to explain why this is a good candidate for a theory with self-triality.

**Some Group Theory: Triality of Spin(8)**

The group Spin(8) has a number of interesting properties. Among these is triality. This is most easily seen in the Dynkin diagram, which has the symmetries of an equilateral triangle, better known as $S_3$:

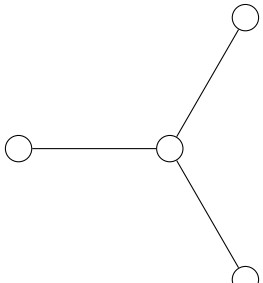

Such symmetries of the Dynkin diagram are associated to *outer automorphisms* of the corresponding Lie algebra and give rise to permutations of certain representations of the corresponding Lie groups. A familiar example is the $\mathbf{Z}_2$ outer automorphism of the $\mathfrak{so}(2r)$ Lie algebra, which acts by permuting the two inequivalent spinor representations of Spin($2r$).

However, the group Spin(8) is special: the vector representation $\mathbf{8_v}$ and the two conjugate spinor representations $\mathbf{8_s}$ and $\mathbf{8_c}$ all have the same dimension, and they are permuted by the $S_3$ triality group.

Note that the group $SO(8)$ does not exhibit triality, in the sense that of the three representations of Spin(8) interchanged by triality, only the vector representation is a true representation of $SO(8)$. On the other hand, $SO(8)/\mathbf{Z}_2$ enjoys triality once more. The simplest way to understand this is to consider the center of Spin(8), which is $\mathbf{Z}_2 \times \mathbf{Z}_2$. The three non-trivial elements in this center can be identified with rotations $(\pi, \pi, \pi, \pi)$, $(\pi, -\pi, -\pi, -\pi)$ and $(2\pi, 0, 0, 0)$ in four orthogonal planes in 8 dimensional space. These are naturally interchanged by triality, with $S_3$ permuting these three elements. This corresponds to the fact that each element of the center acts trivially on exactly one of the three 8 dimensional representations, and as the negative of the identity on the other two. (For example, the vector representation experiences the first two rotations as an overall change of sign, but is left invariant by the final one.) Hence triality acts naturally on Spin(8) or Spin(8)/$(\mathbf{Z}_2 \times \mathbf{Z}_2) = SO(8)/\mathbf{Z}_2$. In this section we focus on the latter; we'll return to the former in Section 3.

## 2.1 Fermions Coupled to a $\mathbf{Z}_2$ Gauge Field

Consider a theory of eight Majorana fermions coupled to a discrete $\mathbf{Z}_2$ gauge field $a$ which acts as $\mathbf{Z}_2 = (-1)^F : \chi_i \mapsto -\chi_i$. We write the action as

$$S = \int d^2x \sum_{i=1}^{8} i \bar{\chi}_i \, \slashed{D}_{a \cdot \rho} \, \chi_i. \tag{2.3}$$

First, we explain the $a \cdot \rho$ notation in the subscript, with $a$ the gauge field and $\rho$ the background spin structure. The spin structure specifies whether the fermions are periodic or anti-periodic around any cycle of the background space. On the Lorentzian manifold $\mathbf{S}^1 \times \mathbf{R}$, we need only specify the boundary conditions around the spatial $\mathbf{S}^1$. On the Euclidean torus $\mathbf{T}^2$, we have two such cycles; taking anti-periodic boundary conditions around the temporal circle computes $\mathrm{Tr}\, e^{-\beta H}$, while periodic boundary conditions computes $\mathrm{Tr}\,(-1)^F e^{-\beta H}$.

Meanwhile, a $\mathbf{Z}_2$ gauge field has a holonomy $\int_\gamma a \in \{0, 1\}$ around any cycle $\gamma$. We can combine this with $\rho$ to form a new spin structure, $a \cdot \rho$. Around any given cycle $\gamma$, the spin structure is unchanged if $\int_\gamma a = 0$, but when $\int_\gamma a = 1$ the boundary conditions are shifted from periodic to anti-periodic and vice-versa.

The fermions $\chi_i$ are taken to transform in the $\mathbf{8_v}$ representation of $SO(8) \subset SO(8)_L \times SO(8)_R$.[1] However, these are not gauge invariant operators. Instead, the faithful continuous global sym-

---

[1]Strictly speaking, this should be $O(8)_L \times O(8)_R$, with the extra $\mathbf{Z}_2$ factors flipping the sign of a single fermion. These $\mathbf{Z}_2$ groups will not play a role in what follows, and we drop them from the discussion to avoid confusion with more various other $\mathbf{Z}_2$ symmetries that will be the focus in what follows.

metry, acting on gauge invariant operators, is $SO(8)/\mathbf{Z}_2$. As explained above, this makes this theory a good candidate to exhibit self-triality. First, we give a flavour of how this works, focussing on the first excited states.

The dynamical gauge field has two effects. First, it projects out states with an odd number of fermionic excitations, $(-1)^F = -1$. Second, it instructs us to sum over all spin structures $\rho$, with the requirement that left- and right-moving fermions experience the same spin structure. Indeed, the theory (2.3) does not depend on the fiducial spin structure $\rho$.

When the fermions have anti-periodic boundary conditions, the first set of excited states sit in the $\mathbf{8_v} \otimes \mathbf{8_v} = \mathbf{1} \oplus \mathbf{28} \oplus \mathbf{35_v}$ representation. Under triality, both the singlet $\mathbf{1}$ and $\mathbf{28}$ are invariant, but the $\mathbf{35_v}$ representation transforms into $\mathbf{35_s}$ and $\mathbf{35_c}$. For triality to hold, states in these representations must occur elsewhere in the spectrum.

To find them, we must turn to the periodic sector. As we reviewed in Section 1, the ground state of each of the left- and right-moving sectors sits in the $\mathbf{8_s} \oplus \mathbf{8_c}$. Of these, one representation is charged under the gauge symmetry $(-1)^F$. It does not matter which, but the charge is the same in both left- and right-moving sectors. We will take $\mathbf{8_c}$ to carry the charge. The ground states in the periodic sector then sit in the representation

$$(\mathbf{8_s} \otimes \mathbf{8_s}) \oplus (\mathbf{8_c} \otimes \mathbf{8_c}). \tag{2.4}$$

These then combine with the $\mathbf{8_v} \otimes \mathbf{8_v}$ states discussed above to give a triality-invariant spectrum as promised. This kind of computation is very familiar to anyone who has computed the spectrum of the superstring; we'll describe the connection in more detail in Section 4.

## 2.2 Bosonization and Triality

The simplest derivation of triality uses bosonization. To our knowledge, this derivation was first presented in [3]. It was also employed in a different context in [16]. In this approach, one starts with fermions transforming in, say, the $\mathbf{8_v}$ representation. These are then exchanged for periodic scalars, which are rearranged and then fermionized back. The net result is a set of fermions transforming in the $\mathbf{8_s}$ or $\mathbf{8_c}$ representation.

As we now explain, bosonization really gives a derivation of the self-triality of the $SO(8)/\mathbf{Z}_2$ theories (2.3) (as opposed to the Spin(8) triality that we will meet in Section 3). First, we pair the fermions, reducing the manifest symmetry from $SO(8) \to U(1)^4$. Each of these complex fermions can be bosonized in favour of a periodic scalar $\theta_\alpha \in [0, 2\pi)$, with $\alpha = 1, 2, 3, 4$, leading to the duality

$$\sum_{i=1}^{8} i\bar{\chi}_i \, \slashed{D}_{a\cdot\rho} \, \chi_i \quad \longleftrightarrow \quad \sum_{\alpha=1}^{4} \left[ \frac{1}{2\pi}(D_{b_\alpha}\theta_\alpha)^2 + i\pi \mathrm{Arf}[b_\alpha \cdot \rho] + i\pi \, a \cup b_\alpha \right]. \tag{2.5}$$

Each of these scalars is charged under a $\mathbf{Z}_2$ gauge symmetry $\theta_\alpha \mapsto \theta_\alpha + \pi$. The associated $\mathbf{Z}_2$ gauge fields are not all independent; the final term above constrains them to obey $\sum_\alpha b_\alpha = 0$.

This is the first time we have met the Arf invariant. This is a mod 2 invariant of a spin structure, and the need to include such a term in the bosonization of a Dirac fermion was pointed out in [15]. On the torus $\mathrm{Arf}[\rho] = 1$ for the RR spin structure and $\mathrm{Arf}[\rho] = 0$ otherwise. In general, on a genus $g$ Riemann surface, $\mathrm{Arf}[\rho] = 1$ for the spin structures that have an odd number of zero modes of the Dirac equation, and vanishes otherwise. We describe a number of properties of the Arf invariant in Appendix A. More details, together with its importance for various 2d dualities, can be found in [15].

In fact, in the present case the Arf invariant is somewhat superfluous. We can eliminate both $a$ and $b_4$ from the scalar theory and use the identity (A.22) to write the duality (2.5) in

the less symmetric form

$$\sum_{i=1}^{8} i\bar{\chi}_i \not{D}_{a\cdot\rho} \chi_i \quad \longleftrightarrow \quad \sum_{\alpha=1}^{3} \frac{1}{2\pi}(D_{b_\alpha}\theta_\alpha)^2 + \frac{1}{2\pi}(\mathcal{D}_{b_1+b_2+b_3}\theta_4)^2 + i\pi \sum_{\alpha<\beta} b_\alpha \cup b_\beta.$$

This makes it explicit that the theory does not depend on the fiducial spin structure $\rho$.

Next, triality: inspired by transformations of the $\mathfrak{so}(8)$ weight lattice, the scalars are replaced by one of two linear combinations,

$$\begin{pmatrix} \theta_1' \\ \theta_2' \\ \theta_3' \\ \theta_4' \end{pmatrix} = \frac{1}{2} \begin{pmatrix} +1 & +1 & +1 & +1 \\ +1 & +1 & -1 & -1 \\ +1 & -1 & +1 & -1 \\ +1 & -1 & -1 & +1 \end{pmatrix} \begin{pmatrix} \theta_1 \\ \theta_2 \\ \theta_3 \\ \theta_4 \end{pmatrix} \quad \text{or} \quad \begin{pmatrix} \theta_1'' \\ \theta_2'' \\ \theta_3'' \\ \theta_4'' \end{pmatrix} = \frac{1}{2} \begin{pmatrix} +1 & +1 & +1 & -1 \\ +1 & +1 & -1 & +1 \\ +1 & -1 & +1 & +1 \\ +1 & -1 & -1 & -1 \end{pmatrix} \begin{pmatrix} \theta_1 \\ \theta_2 \\ \theta_3 \\ \theta_4 \end{pmatrix}.$$

(2.6)

Note that if $\theta_\alpha$ were $2\pi$ periodic variables, without any identifications, then $\theta_\alpha'$ and $\theta_\alpha''$ would fail to be. For example, $\theta_1 \to \theta_1 + 2\pi$ shifts $\theta_I'$ and $\theta_I''$ by $\pi$. However, the presence of various $\mathbf{Z}_2$ quotients conspires to guarantee that the theories before and after this transformation are the same. Explicitly, the above theory describes 4 bosons on the $SO(8)/\mathbf{Z}_2$ lattice defined by four identifications $(\theta_I, \theta_4) \to (\theta_I + \pi, \theta_4 + \pi)$ for $I = 1, 2, 3$ and $\theta_4 \to \theta_4 + 2\pi$. The lattice is left invariant by both of the above transformations.

One can then refermionize the new bosonic fields, reversing the above process, to give two equivalent theories, $\chi'$ and $\chi''$, each coupled to a $\mathbf{Z}_2$ gauge field. These new fermions now transform under $\mathbf{8_s}$ and $\mathbf{8_c}$ respectively. To see this, note that we can continuously deform $\theta_1 \to \theta_1 + 2\pi$ which, as we mentioned above, corresponds to $\theta_I' \to \theta_I' + \pi$ and $\theta_I'' \to \theta_I'' + \pi$. This, in turn, means that the new fermionic fields $\chi'$ and $\chi''$ change sign under this transformation: they are either $\mathbf{8_s}$ or $\mathbf{8_c}$ fields. By considering $\theta_I \to \theta_I + \pi$, under which only $\chi$ and $\chi''$ change sign, we can conclude that we get one each of $\mathbf{8_s}$ and $\mathbf{8_c}$.

This establishes self-triality of the $SO(8)/\mathbf{Z}_2$ theory, written schematically as

$$\sum_{i=1}^{8} i\bar{\chi}_i \not{D}_{a\cdot\rho}^{\mathbf{v}} \chi_i \quad \longleftrightarrow \quad \sum_{i=1}^{8} i\bar{\chi}_i' \not{D}_{a\cdot\rho}^{\mathbf{s}} \chi_i' \quad \longleftrightarrow \quad \sum_{i=1}^{8} i\bar{\chi}_i'' \not{D}_{a\cdot\rho}^{\mathbf{c}} \chi_i'',$$

(2.7)

where the superscripts $\mathbf{v}$, $\mathbf{s}$ and $\mathbf{c}$ denote the Spin(8) representations of the corresponding fermions. Although we have discussed the duality on a torus, it holds on a general Riemann surface.

This theory is familiar in the context of conformal field theory, where it is described by the $\widehat{\mathfrak{so}(8)}_1$ WZW model. (See, for example, [17].) There are four integrable representations of this affine algebra, corresponding to $\mathbf{1}$, $\mathbf{8_v}$, $\mathbf{8_s}$ and $\mathbf{8_c}$, denoted as $\hat{\omega}_i$. The partition function of the $SO(8)/\mathbf{Z}_2$ theory is simply $Z = \sum_i |\chi_{\hat{\omega}_i}|^2$, the sum of the characters of these four representations, and triality permutes the three non-vacuum terms.

Each of the theories above also enjoys a number of global $\mathbf{Z}_2$ symmetries. Of particular interest is the chiral symmetry

$$\mathbf{Z}_2^A : \chi_i \mapsto \gamma^3 \chi_i.$$

Insisting that the chiral symmetry is preserved would seem to protect the fermions $\chi_i$ from getting a mass, since $\mathbf{Z}_2^A : \bar{\chi}_i \chi_i \mapsto -\bar{\chi}_i \chi_i$. The study of interacting topological insulators shows that it is, nonetheless, possible for the fermions $\chi_i$ to become gapped without breaking $\mathbf{Z}_2^A$ [5,6]. The key to understanding how this is possible is to track the action of $\mathbf{Z}_2^A$ through the different trialities. Before doing this, it will prove useful to first construct a closely related triality that acts only on Majorana-Weyl fermions. We will then return to our $SO(8)/Z_2$ theory in Section 3.2, and to the study of interactions in Section 5.

# 3 Chiral Triality

The purpose of this section is to write down a duality in which one of the theories consists only of free fermions. In fact, it turns out to be simplest to do this for a chiral theory, consisting of 8 left-moving Majorana-Weyl fermions and then use this to construct trialities with both left- and right-moving fermions. We will then revisit the $SO(8)/\mathbf{Z}_2$ theory in Section 3.2, using our new knowledge to track the various global $\mathbf{Z}_2$ symmetries through the triality chain.

## 3.1 Partition Functions

The simplest way to construct such dualities is to compare the partition functions on a torus. We introduce the usual modular parameter $\tau$ on the torus and its exponent

$$q = e^{2\pi i \tau}.$$

We start by ignoring fugacities for the $SO(8)$ symmetry. In their absence, the computation of the partition functions for a chiral Majorana fermion is standard textbook material in conformal field theory. (See, for example [17], or the lecture notes [18].) There is a separate partition function for each of the four spin structures, $\rho$ and we denote these as, for example, $_A\boxed{\phantom{x}}_P$ for the case of a periodic spatial structure and an anti-periodic time structure. The partition functions for 8 Majorana-Weyl fermions are

$$
{}_A\boxed{\phantom{x}}_A = q^{-1/6} \prod_{n=0}^{\infty} (1 + q^{n+1/2})^8 = \frac{\vartheta_3^4}{\eta^4},
$$

$$
{}_P\boxed{\phantom{x}}_A = q^{-1/6} \prod_{n=0}^{\infty} (1 - q^{n+1/2})^8 = \frac{\vartheta_4^4}{\eta^4},
$$

$$
{}_A\boxed{\phantom{x}}_P = \frac{1}{2^4} q^{1/3} \prod_{n=0}^{\infty} (1 + q^n)^8 = \frac{\vartheta_2^4}{\eta^4},
$$

with $\eta = q^{1/24} \prod_{n=1}^{\infty}(1 - q^n)$ the Dedekind eta function and $\vartheta_i$ the Jacobi theta functions. Finally, the when $\rho = PP$ there is a zero mode and we have

$$
{}_P\boxed{\phantom{x}}_P = 0.
$$

These functions obey the well-known identity

$$
{}_A\boxed{\phantom{x}}_A - {}_P\boxed{\phantom{x}}_A - {}_A\boxed{\phantom{x}}_P = \frac{1}{\eta^4}\left(\vartheta_3^4 - \vartheta_4^4 - \vartheta_2^4\right) = 0.
$$

For our purposes, this is better written as

$$
{}_A\boxed{\phantom{x}}_A = \frac{1}{2}\left( {}_A\boxed{\phantom{x}}_A + {}_A\boxed{\phantom{x}}_P + {}_P\boxed{\phantom{x}}_A \pm {}_P\boxed{\phantom{x}}_P \right). \tag{3.8}
$$

This is a duality: the left-hand side describes 8 free fermions, with $\rho = AA$ boundary conditions. The equality says that this is equivalent to 8 fermions coupled to a $\mathbf{Z}_2$ gauge field which sums over spin structures. The choice of sign corresponds to the choice of a topological term for the gauge field, as we explain below.

In fact, the equality (3.8) is a manifestation of triality, with the three theories (including the choice of sign on the right-hand side) containing fermions transforming in different representations of the Spin(8) global symmetry. To see this, we need to generalize (3.8) to include fugacities for the Spin(8) flavour symmetry. At the same time, we would like to understand how to write an analogous identity when the free fermions experience other spin structures.

**Adding Fugacities**

We first define the partition function for fermions transforming in the $\mathbf{8_v}$. We introduce fugacities $z_i$, $i = 1, 2, 3, 4$ for the $U(1)^4 \subset \mathrm{Spin}(8)$ global symmetry.[2]

We will use the notation $\rho_\alpha = 0$ for anti-periodic boundary conditions and $\rho_\alpha = 1$ for periodic boundary conditions around the temporal ($\alpha = 0$) and spatial ($\alpha = 1$) circles. For anti-periodic boundary conditions on the spatial circle, we have $\rho_1 = 0$ and the partition function

$$\Theta_v[\rho; q, z] = q^{-1/6} \prod_{\lambda \in \mathbf{8_v}} \prod_{n=0}^{\infty} \left(1 + (-1)^{\rho_0} z^\lambda q^{n+1/2}\right). \tag{3.9}$$

When we set $z = 1$, this result coincides with the previous expressions $_A\boxed{\phantom{x}}_A$ and $_P\boxed{\phantom{x}}_A$.

The novelty is in the fugacities. Here the product is over the eight weights $\lambda$ in the $\mathbf{8_v}$ weight system, and we have introduced the notation $z^\lambda = \prod_{i=1}^{4} z_i^{\lambda_i}$. So, for example, the eight possible values of $z^\lambda$ with $\lambda$ in the $\mathbf{8_v}$ weight system are

$$z^\lambda \in \left\{ z_1, \frac{1}{z_1}, \frac{z_2}{z_1}, \frac{z_1}{z_2}, \frac{z_3 z_4}{z_2}, \frac{z_2}{z_3 z_4}, \frac{z_4}{z_3}, \frac{z_3}{z_4} \right\}. \tag{3.10}$$

In what follows, we will need similar expressions for $z^\lambda$ where $\lambda$ now sit in either the $\mathbf{8_s}$ or $\mathbf{8_c}$ weight system. These arise as follows: permuting the representations $\mathbf{8_v} \to \mathbf{8_s} \to \mathbf{8_c} \to \mathbf{8_v}$ is implemented by similarly permuting $z_1 \to z_3 \to z_4 \to z_1$. Meanwhile, exchanging $z_3 \leftrightarrow z_4$ corresponds to spinor conjugation $\mathbf{8_s} \leftrightarrow \mathbf{8_c}$.

For periodic boundary conditions on the spatial circle, so $\rho_1 = 1$, the ground states sit in the $\mathbf{8_s}$ and $\mathbf{8_c}$ representations and the partition function is given by

$$\Theta_v[\rho; q, z] = q^{1/3} \left[ \sum_{\lambda \in \mathbf{8_s}} z^\lambda + (-1)^{\rho_0} \sum_{\lambda \in \mathbf{8_c}} z^\lambda \right] \prod_{\lambda \in \mathbf{8_v}} \prod_{n=1}^{\infty} \left(1 + (-1)^{\rho_0} z^\lambda q^n\right). \tag{3.11}$$

When we set $z = 1$, this result coincides with the previous expressions $_A\boxed{\phantom{x}}_P$ and $_P\boxed{\phantom{x}}_P$.

Note that we have made a choice in constructing this partition function: we have assigned the $\mathbf{8_s}$ vacua fermion number $(-1)^F = +1$ and the $\mathbf{8_c}$ vacua fermion number $(-1)^F = -1$.

We could alternatively pick the fundamental fermions to transform in the $\mathbf{8_s}$ or $\mathbf{8_c}$ representations. For each of these, we have a corresponding partition function $\Theta_s[\rho; q, z]$ and $\Theta_c[\rho; q, z]$. Importantly, our convention for the assignment of $(-1)^F$ charge for the ground states preserves the above cyclic symmetry. This means that for $\rho_1 = 1$ we have

$$\Theta_s[\rho; q, z] = q^{1/3} \left[ \sum_{\lambda \in \mathbf{8_c}} z^\lambda + (-1)^{\rho_0} \sum_{\lambda \in \mathbf{8_v}} z^\lambda \right] \prod_{\lambda \in \mathbf{8_s}} \prod_{n=0}^{\infty} \left(1 + (-1)^{\rho_0} z^\lambda q^n\right) \tag{3.12}$$

and

$$\Theta_c[\rho; q, z] = q^{1/3} \left[ \sum_{\lambda \in \mathbf{8_v}} z^\lambda + (-1)^{\rho_0} \sum_{\lambda \in \mathbf{8_s}} z^\lambda \right] \prod_{\lambda \in \mathbf{8_c}} \prod_{n=0}^{\infty} \left(1 + (-1)^{\rho_0} z^\lambda q^n\right). \tag{3.13}$$

---

[2]The partition functions described in this section are the $\mathbf{Z}_2$ Fourier transforms of the characters of $\widehat{\mathfrak{so}(8)}_1$.

With these partition functions in hand, we can now describe how triality acts on free fermions. The partition functions (3.9), (3.11), (3.12) and (3.13) obey the following identity:

$$
\begin{aligned}
\Theta_v[A \cdot \rho_{NS}; q, z] &= \frac{1}{2} \sum_{a_0, a_1 = 0}^{1} (-1)^{a \cup A + \mathrm{Arf}[a \cdot \rho_{NS}]} \Theta_s[a \cdot \rho_{NS}; q, z] \\
&= \frac{1}{2} \sum_{a_0, a_1 = 0}^{1} (-1)^{a \cup A + \mathrm{Arf}[A \cdot \rho_{NS}]} \Theta_c[a \cdot \rho_{NS}; q, z]
\end{aligned}
\tag{3.14}
$$

where we are adopting the convention that lower case $\mathbf{Z}_2$ gauge fields like $a$ are dynamical, and so summed over in the partition function, while upper case $\mathbf{Z}_2$ gauge fields like $A$ are background. In this, and further expressions, the cup product is short-hand for the integral $\int a \cup A$.

Just like the simpler version of triality (3.8), the free fermions most naturally sit on a privileged background spin structure $\rho_{NS} = AA$. But we have now introduced a background gauge $A$ which can use to shift the spin structure experienced by the free fermions to $A \cdot \rho_{NS}$.

The theories with $\mathbf{8_s}$ and $\mathbf{8_c}$ fermions both have a dynamical $\mathbf{Z}_2$ gauge field $a$ which acts on the fermions as $(-1)^F$. The terms $a \cup A$ and $\mathrm{Arf}[a \cdot \rho_{NS}]$ terms dictate the charges of different states. Specifically, the $a \cup A$ term gives an extra $\mathbf{Z}_2$ gauge charge to states when $A_1 = 1$. Similarly, the $\mathrm{Arf}[a \cdot \rho_{NS}]$ term gives an extra $\mathbf{Z}_2$ gauge charge to states when $a_1 = 1$.

Details of the matching of the spectra for low-lying states can be found in Appendix B.1.

## 3.2 Trialities with Left- and Right-Movers

To keep the notation simple, we will omit the $q$ and $z$ fugacities in the argument of the partition function, and write the first duality in (3.14) as

$$
\begin{aligned}
\Theta_v[A \cdot \rho_{NS}] &= \frac{1}{2} \sum_{a} (-1)^{a \cup A + \mathrm{Arf}[a \cdot \rho_{NS}]} \Theta_s[a \cdot \rho_{NS}] \\
&= \frac{1}{2} \sum_{a} (-1)^{a \cup A + \mathrm{Arf}[A \cdot \rho_{NS}]} \Theta_c[a \cdot \rho_{NS}].
\end{aligned}
\tag{3.15}
$$

For theories with both left- and right-moving fermions, we take the product of the partition function with its conjugate. To this end, we define

$$
\mathcal{Z}_v[V, A; \rho] = \overline{\Theta_v[(V + A) \cdot \rho]} \, \Theta_v[V \cdot \rho],
\tag{3.16}
$$

with similar expressions for $\mathcal{Z}_s$ and $\mathcal{Z}_c$. The background $\mathbf{Z}_2$ gauge fields $V$ and $A$ have been constructed so that they couple to the vector and axial symmetries of the free fermions,

$$
\mathbf{Z}_2^V : \chi_i \mapsto -\chi_i \quad \text{and} \quad \mathbf{Z}_2^A : \chi_i \mapsto -\gamma^3 \chi_i.
$$

It is then simple to use the triality (3.15) to derive a triality between theories with both left- and right-movers. Using the property of the Arf invariant (A.22), we have

$$
\begin{aligned}
\mathcal{Z}_v[V, A; \rho_{NS}] &= \frac{1}{4} \sum_{a, b} (-1)^{a \cup V + b \cup (V + A) + \mathrm{Arf}[a \cdot \rho_{NS}] + \mathrm{Arf}[b \cdot \rho_{NS}]} \, \mathcal{Z}_s[a, a + b; \rho_{NS}] \\
&= \frac{1}{4} \sum_{a, b} (-1)^{a \cup V + b \cup (V + A) + A \cup V + \mathrm{Arf}[A \cdot \rho_{NS}]} \, \mathcal{Z}_c[a, a + b; \rho_{NS}].
\end{aligned}
\tag{3.17}
$$

This duality tells us that 8 free Majorana fermions are dual to 8 Majoranas coupled to a $\mathbf{Z}_2 \times \mathbf{Z}_2$ chiral gauge field. The background gauge fields $V$ and $A$ allow us track the action of the global chiral symmetries through the duality. On the right-hand-side, these couple to the kink states, or disorder operators, that come from the twisted sector of the $a$ and $b$ gauge fields.

### $SO(8)/\mathbf{Z}_2$ Self-Triality Revisited

Given a duality, it is often possible to construct new dualities by gauging symmetries on both sides. In $d = 1 + 1$ dimensions, it is particularly natural to gauge $\mathbf{Z}_2$ symmetries, a procedure which is sometimes referred to as "orbifolding". In this case, a new "quantum" $\mathbf{Z}_2$ global symmetry emerges in the new theory allowing one to repeat the procedure. For simple theories, this leads to a duality web in $d = 1 + 1$ dimensions relating, for example, bosonization with Kramers-Wannier duality [10–13,15]. Aspects of this orbifolding procedure in supersymmetric dualities were discussed, for example, in [19,20].

It is straightforward to do this gauging in the present case. We start with the partition function $\mathcal{Z}_v$ defined in (3.16) and gauge the vector-like symmetry $\mathbf{Z}_2$, to define

$$\mathcal{Z}_{v/\mathbf{Z}_2}[V,A;\rho_{NS}] = \frac{1}{2}\sum_a (-1)^{a\cup V + \mathrm{Arf}[(V+A)\cdot\rho_{NS}]}\,\mathcal{Z}_v[a,A;\rho_{NS}].$$

We define $\mathcal{Z}_{s/\mathbf{Z}_2}$ and $\mathcal{Z}_{c/\mathbf{Z}_2}$ in the same way. In each of these, the background gauge field $V$ is the "quantum" $\mathbf{Z}_2$ symmetry that, through the coupling $a \cup V$, measures the charge of disorder operators in the dynamical gauge field. We have also dressed the partition function $\mathcal{Z}_{v/\mathbf{Z}_2}$ with Arf terms for the background fields.

Now we use this to derive a new duality. We gauge the $\mathbf{Z}_2^V$ symmetry on both sides of the duality (3.17). The left-hand-side becomes $\mathcal{Z}_{v/\mathbf{Z}_2}$ defined above. Meanwhile, on the right-hand-side, promoting $V$ to a dynamical gauge field acts as a Lagrange multiplier and relates the two original gauge fields $a$ and $b$. A short calculation reveals the triality

$$\mathcal{Z}_{v/\mathbf{Z}_2}[V,A;\rho_{NS}] = \mathcal{Z}_{s/\mathbf{Z}_2}[V+A,V;\rho_{NS}] = \mathcal{Z}_{c/\mathbf{Z}_2}[A,V+A;\rho_{NS}]. \tag{3.18}$$

If we set $V = A = 0$, this reduces to our earlier $SO(8)/\mathbf{Z}_2$ triality described in Section 2. The advantage of the partition function approach is that we can straightforwardly track the action of the $\mathbf{Z}_2^V$ and $\mathbf{Z}_2^A$ global symmetries through the triality.

Indeed, there is something of a surprise waiting for us: the triality (3.18) says that the $\mathbf{Z}_2^V$ symmetry, which is non-chiral in the first theory, becomes a chiral transformation in the second, where it acts as $\mathbf{Z}_2^V : \chi'_i \mapsto -\gamma^3\chi'_i$. We'll explore the implications of this in Section 5.

The self-triality (3.18) only exhibits $\mathbf{Z}_3$ cyclic permutations, a subgroup of the full triality group $S_3$. In Appendix B.2 we describe how the full triality group acts on the theories, and how it is intertwined with time reversal. We also provide some comments on the extension to general spin structures.

## 4 The Superstring

Triality plays an important role in the quantization of the superstring. This is standard textbook material which can be found, for example, in [21]. Here we review this story, focussing on the aspects which touch upon the trialities described above.

The $d = 1 + 1$ dimensional worldsheet theory of the string enjoys diffeomorphism invariance. This means that any matter on the string is forbidden from having a gravitational anomaly. In particular, if the worldsheet is a torus then the theory living on the worldsheet must be modular invariant.

In light-cone gauge, the superstring reduces to a theory of free bosons, together with 8 Majorana fermions. There are two ways to render such a theory modular invariant. The first is to restrict to the $\rho = PP$ (or Ramond-Ramond) spin structure for the fermions; this has the property that it transforms into itself under modular transformations. This approach is called the Green-Schwarz string and has the added benefit that theory exhibits a manifest spacetime (i.e. $d = 9 + 1$) supersymmetry; such theories go by the name of Type II string theory.

The second approach, known as the Ramond-Neveu-Schwarz, or RNS string, is to sum over spin structures and subsequently throw away certain states in a consistent manner, a procedure known as the GSO projection [8].

## 4.1 Type II Theories

In the context of string theory, the Spin(8) global symmetry on the worldsheet is to be thought of as a rotation symmetry in $d = 9 + 1$ dimensional spacetime, Spin(8) $\subset$ Spin(9, 1). For this reason, it is best to view the free fermions on the worldsheet as transforming in one of the two spinor representation of of Spin(8), reflecting the fact that they are associated to fermions in the larger spacetime.

There is a straightforward cyclic permutation of the triality (3.14), $\mathbf{8_v} \to \mathbf{8_s} \to \mathbf{8_c}$. This allows us to relate free Majorana-Weyl fermions transforming in the $\mathbf{8_s}$ or $\mathbf{8_c}$ to fermions in the $\mathbf{8_v}$ coupled to a $\mathbf{Z}_2$ gauge field, resulting in an equality of partition functions analogous to (3.15):

$$\Theta_s[A \cdot \rho_{NS}] = \frac{1}{2} \sum_a (-1)^{a \cup A + \text{Arf}[A \cdot \rho_{NS}]} \Theta_v[a \cdot \rho_{NS}],$$

$$\Theta_c[A \cdot \rho_{NS}] = \frac{1}{2} \sum_a (-1)^{a \cup A + \text{Arf}[a \cdot \rho_{NS}]} \Theta_v[a \cdot \rho_{NS}].$$

Note that only the second of these includes an Arf term for the dynamical gauge field.

As we mentioned above, the Green-Schwarz string involves free fermions on a Ramond-Ramond spin structure. We can achieve this by taking $A_0 = A_1 = 1$. We then define $\rho_R = A \cdot \rho_{NS}$, to be the spin structure that is periodic on both cycles. Shifting $a \to a + A$, we have

$$\Theta_s[\rho_R] = -\frac{1}{2} \sum_a (-1)^{a \cup A} \Theta_v[a \cdot \rho_R],$$

$$\Theta_c[\rho_R] = \frac{1}{2} \sum_a (-1)^{a \cup A + \text{Arf}[a \cdot \rho_R]} \Theta_v[a \cdot \rho_R]. \tag{4.19}$$

Note that the Arf$[a \cdot \rho_R]$ term on the right-hand-side determines whether we keep the $\mathbf{8_s}$ or $\mathbf{8_c}$ ground state in the periodic sector.

The presence of the fixed $A = (1, 1)$ term on the right-hand-side ensures that the ground state in the anti-periodic sector carries $\mathbf{Z}_2$ charge and so is projected out. From the field theory perspective, it is slightly unusual to project out the vacuum in this manner, leaving behind the excited states. However, it is very familiar from the perspective of string theory since this is how the tachyon is removed.

To construct modular invariant theories, we need to put together left- and right-moving fermions. There are two ways to achieve this, known as the Type IIA and Type IIB string respectively. They are

$$\mathcal{Z}_{IIA} = \overline{\Theta_s[\rho_R]} \Theta_c[\rho_R] = -\frac{1}{4} \sum_{a,b} (-1)^{(a+b) \cup A + \text{Arf}[a \cdot \rho_R]} \overline{\Theta_v[b \cdot \rho_R]} \Theta_v[a \cdot \rho_R]$$

and

$$\mathcal{Z}_{IIB} = \overline{\Theta_s[\rho_R]} \Theta_s[\rho_R] = \frac{1}{4} \sum_{a,b} (-1)^{(a+b) \cup A} \overline{\Theta_v[b \cdot \rho_R]} \Theta_v[a \cdot \rho_R].$$

In each of these expressions, the first equality describes the Green-Schwarz string, while the second describes the RNS string, with the two chiral gauge fields $a$ and $b$ implementing the GSO projection.

A comment: it may look strange that the left-hand side of (4.19) is modular invariant, while the right-hand-side needs a fixed, background field $A = (1, 1)$. This reflects the fact that we made a particular choice of phases when defining the chiral partition functions in (3.9) and (3.11). On the left-hand side, these choices of phase cancel each other out, but on the right-hand side this is not true as left- and right-movers experience different spin structures. Instead, the choice of phase is cancelled by the phase of the background term involving $A$.

## 4.2 Type 0 Theories

There are other superstring theories which are consistent in the sense that they have modular invariant worldsheets, but do not result in spacetime supersymmetry. In particular, this means that the spacetime theory has a tachyon and so is unstable. These theories, first described in [22], go by the name of Type 0 theories.

In fact, we have already met Type 0B theory. This is precisely the $SO(8)/\mathbf{Z}_2$ theory that we first discussed in Section 2 and elaborated upon in Section 3.2. The fact that we have just a single $\mathbf{Z}_2$ gauge field, ensures that the left- and right-handed fermions experience the same spin structure, which the defining feature of the type 0 theories. As we saw in (2.4), the Ramond-Ramond ground states sit in the $(\mathbf{8_s} \otimes \mathbf{8_s}) \oplus (\mathbf{8_c} \otimes \mathbf{8_c})$ representation, which identifies this theory as Type 0B.

It is not difficult to construct the Type 0A theory and its triality properties. We return to the duality between the free fermions and $\mathbf{Z}_2 \times \mathbf{Z}_2$ chiral gauge theory (3.17). We saw in Section 3.2 that gauging the background $\mathbf{Z}_2^V$ symmetry results in the $SO(8)/\mathbf{Z}_2$ triality that we identify as Type 0B. We can repeat this procedure, but this time introduce an $\mathrm{Arf}[V \cdot \rho]$ term before we promote $V$ to a dynamical gauge field. The result is the triality

$$
\begin{aligned}
& \frac{1}{2} \sum_a (-1)^{a \cup V + \mathrm{Arf}[a \cdot \rho_{NS}]} \, \mathcal{Z}_v[a, A; \rho_{NS}] \\
= \; & \frac{1}{4} \sum_{a,b} (-1)^{a \cup V + b \cup (A+V) + a \cup b + \mathrm{Arf}[V \cdot \rho_{NS}]} \, \mathcal{Z}_s[a, a+b; \rho_{NS}] \\
= \; & \frac{1}{4} \sum_{a,b} (-1)^{a \cup (A+V) + b \cup V + A \cup V + \mathrm{Arf}[(a+b) \cdot \rho_{NS}] + \mathrm{Arf}[V \cdot \rho_{NS}]} \, \mathcal{Z}_c[a, a+b; \rho_{NS}].
\end{aligned}
$$

This is less symmetric than our other trialities; it relates a $\mathbf{Z}_2$ gauge theory coupled to the $\mathbf{8_v}$ fermions to $\mathbf{Z}_2 \times \mathbf{Z}_2$ chiral gauge theories coupled to either $\mathbf{8_s}$ or $\mathbf{8_c}$ fermions. The presence of the $\mathrm{Arf}[a \cdot \rho_{NS}]$ term in the first line means that, in the Ramond-Ramond sector, the gauge symmetry now projects onto states with $(-1)^F = -1$ rather than $+1$. The ground states in this sector then lie in the $(\mathbf{8_s} \otimes \mathbf{8_c}) \oplus (\mathbf{8_c} \otimes \mathbf{8_s})$ representation, which is the hallmark of the Type 0A string.

# 5 Symmetric Mass Generation

In this final section, we turn to a rather different application of triality: the question of when fermions can gain a mass.

It is a basic fact of free quantum field theory that massless fermions enjoy more symmetries than massive fermions. A fundamental question is whether this is an artefact of working around the Gaussian fixed point. The idea that it may be possible to give fermions masses without breaking certain protective symmetries goes by the name of *symmetric mass generation*. There has been a great deal of work exploring this possibility in various situations [4–7, 9, 23–30]

There are two stories of symmetric mass generation that involve Majorana fermions in $d = 1+1$ dimensions. The first, due to Fidkowski and Kitaev [4], starts by giving $N_f$ fermions a mass $m$. This preserves a time reversal symmetry obeying $T^2 = 1$. There are two phases of the theory, characterized by the sign of $m$, and a domain wall interpolating between these two phases exhibits gapless Majorana zero modes, protected by the time reversal symmetry. Fidkowski and Kitaev showed that it is possible to gap these zero modes, while preserving time reversal symmetry, only when $N_f$ is a multiple of 8.

Here we focus on a second, closely related story, in which the $d = 1+1$ Majorana fermions are gapless. In the context of SPT phases, these fermions can be viewed as edge modes of gapped $d = 2+1$ dimensional theory, although this is not necessary in what follows. The question we would like to address is: what symmetries are broken if the fermions get a mass? Once again, something special happens for 8 Majorana fermions [5–7, 9].

**Symmetric Mass Generation in** $d = 1+1$

We will care about the two, discrete global symmetries

$$\mathbf{Z}_2^V \times \mathbf{Z}_2^A.$$

The result of [4–6] is that it is possible for 8 Majorana fermions to get a mass preserving both of these symmetries. In what follows, we review this result and see how it arises from an interplay of gauge symmetry and the Arf invariant.

To put the result in context, let's first review some basic facts about fermions in $d = 1+1$. Suppose that we have free fermions which we take to transform in the $\mathbf{8_v}$. If we give them a mass, clearly the $\mathbf{Z}_2^A$ symmetry is explicitly broken, since

$$\mathbf{Z}_2^A : \bar{\chi}_i \chi_i \;\mapsto\; -\bar{\chi}_i \chi_i.$$

It is, of course, no surprise that fermion masses explicitly break the discrete chiral symmetry.

In $d = 1+1$ dimensions, it is sometimes possible to give fermions a mass in a way that preserves the axial symmetry. This arises, for example, if an even number of fermions are coupled to a $\mathbf{Z}_2$ gauge field. (An even number is needed to ensure that $\mathbf{Z}_2^A$ is non-anomalous.) In this case, disorder operators can gap the system, explicitly breaking a quantum $\mathbf{Z}_2^V$ symmetry but preserving $\mathbf{Z}_2^A$. This also has a simple description in the bosonized language, where we can add a relevant deformation for either the periodic scalar or the dual scalar.

However, 8 Majorana fermions are special. It turns out that it is possible to give 8 fermions a mass, without introducing dynamical gauge fields, while preserving *both* $\mathbf{Z}_2^A$ and $\mathbf{Z}_2^V$. Here we review the original derivation of this result which leans heavily on the existence of the Spin(8) triality [5,6]. Closely related approaches include understanding the modular properties of the partition function in the presence of $\mathbf{Z}_2^A \times \mathbf{Z}_2^V$ gauge fields [31], and a study of the braiding statistics of the $d = 2+1$ SPT upon gauging these discrete symmetries [32].

## 5.1 Interactions Preserving Spin(8)

The interactions that we will need are the familiar four-fermion terms, first introduced by Gross and Neveu [33]. We start by reviewing the physics of these interactions when we preserve the full Spin(8) global symmetry group. Ultimately, we will be interested in a set of interactions that preserve only Spin(7); we discuss these in Section 5.2.

We start by reviewing the physics of the $SO(8)$ Gross-Neveu model. For this purpose, we add an interaction term for the $\mathbf{8_v}$ fermions,

$$\mathcal{L}_{SO(8)} = -A \left( \sum_{i=1}^{8} \bar{\chi}_i \chi_i \right)^2. \tag{5.20}$$

The physics of this is well known. The theory is asymptotically free and the coupling $A$ runs, to be replaced by a strong coupling scale $\Lambda \sim \mu e^{-16\pi/A}$. At this scale, the fermion bilinear develops an expectation value

$$\left\langle \sum_{i=1}^{8} \bar{\chi}_i \chi_i \right\rangle = \pm \Lambda.$$

This spontaneously breaks the $\mathbf{Z}_2^A$ symmetry, resulting in two, gapped ground states.

The kink which interpolates between the two ground states has 8 Majorana zero modes. Quantising these shows that the kinks transform in the $\mathbf{8_s} \oplus \mathbf{8_c}$ representation of the Spin(8) global symmetry.

We see that the $SO(8)$ Gross-Neveu coupling does nothing to help us with symmetric mass generation: although the original interaction preserves all symmetries, the $\mathbf{Z}_2^A$ discrete chiral symmetry is spontaneously broken, allowing the fermions to get a mass.

Suppose, instead, that we add the $SO(8)$ Gross-Neveu interaction (5.20) for the $\mathbf{8_s}$ fermions. Recall from the duality (3.17) tells that these $\mathbf{8_s}$ fermions are charged under a $\mathbf{Z}_2 \times \mathbf{Z}_2$ gauge symmetry, with the action of various symmetries encoded in the partition function

$$\mathcal{Z}_v[V,A;\rho_{NS}] = \frac{1}{4} \sum_{a,b} (-1)^{a\cup V + b\cup(V+A) + \mathrm{Arf}[a\cdot\rho_{NS}] + \mathrm{Arf}[b\cdot\rho_{NS}]} \, \mathcal{Z}_s[a, a+b; \rho_{NS}].$$

The dynamics is the same, with the $\mathbf{8_s}$ fermions developing an expectation value

$$\left\langle \sum_{i=1}^{8} \bar{\chi}'_i \chi'_i \right\rangle = \Lambda.$$

This now "spontaneously breaks" the chiral gauge symmetry $\mathbf{Z}_2^{a+b}$. The fact that we have broken a gauge symmetry, rather than a global symmetry, means that this time there is – at least for now – just a single ground state, rather than the two ground states we saw for the $\mathbf{8_v}$ fermions.

Nonetheless, we do not have to look far to see the two ground states re-emerging. These arise from the $\mathbf{Z}_2^a$ gauge symmetry, which survives unscathed. As in the previous case, the fermions become gapped and we may integrate them out. We set $a + b = 0$, reflecting the fact that $\mathbf{Z}_2^{a+b}$ is broken, leaving us with the topological action

$$S_{\mathbf{Z}_2} = i\pi a \cup A. \tag{5.21}$$

The $\mathbf{Z}_2$ gauge theory has two ground states, corresponding to the two holonomies for the gauge field around the spatial circle. The action (5.21) is telling us that these two ground states carry different $\mathbf{Z}_2^A$ charges. Another way of saying this is that the linear combinations $|a_1 = 0\rangle \pm |a_1 = 1\rangle$ are exchanged by $\mathbf{Z}_2^A$: in other words, the global $\mathbf{Z}_2^A$ symmetry is spontaneously broken. Meanwhile, $\mathbf{Z}_2^V$ remains unbroken. This matches the symmetry-breaking pattern of the triality-dual $\mathbf{8_v}$ theory, as it must since the two Gross-Neveu terms we added are actually dual to each other.

The upshot of this analysis is that the $SO(8)$ Gross-Neveu deformation gives rise to a gapped theory in which only the $\mathbf{Z}_2^V$ global symmetry is preserved, but $\mathbf{Z}_2^A$ is spontaneously broken.

## 5.2 Interactions Preserving Spin(7)

To exhibit symmetric mass generation, we instead need to study an interaction that preserves just an Spin(7) $\subset$ Spin(8) subgroup of the full global symmetry. The field theory analysis of this interaction was first performed by Fidkowski and Kitaev [4], and this was re-purposed for

$\mathbf{Z}_2^V \times \mathbf{Z}_2^A$ symmetric mass generation in [5, 6]. The fact that the interactions should preserve Spin(7), rather than $SO(7)$, was stressed in [34].

We will consider the $\mathbf{8_s}$ fermions, and add the interactions

$$\mathcal{L}_{SO(7)} = -A\left(\sum_{i=1}^{7} \bar{\chi}'_i \chi'_i\right)^2 - B\left(\sum_{i=1}^{7} \bar{\chi}'_i \chi'_i\right)\bar{\chi}'_8 \chi'_8.$$

Following [4], we set $A \gg B$. This means that we first focus on the dynamics of the $SO(7)$ Gross-Neveu coupling. Once again, the theory flows to a strong coupling scale $\Lambda$ and develops an expectation value

$$\left\langle \sum_{i=1}^{7} \bar{\chi}'_i \chi'_i \right\rangle = \Lambda.$$

As before, this breaks the $\mathbf{Z}_2^{a+b}$ chiral gauge group, but leaves the $\mathbf{Z}_2^a$ unbroken.

The 7 Majorana fermions obtain a mass $\sim \Lambda$. Without loss of generality, these can be taken to sit in the topologically trivial phase and can be simply integrated out, resulting in an effective action for the remaining, eighth Majorana fermion:

$$S_{\text{Maj}} = i\bar{\chi}_8 \,\slashed{\partial}_{a \cdot \rho_{NS}}\, \chi_8 - m\bar{\chi}_8 \chi_8 + a \cup A.$$

Here, the mass is given by $m = B\Lambda$. Note that the gauge field $V$ is absent from this effective action. This is the field that couples to fermion number $(-1)^F$ in the free fermion side of the triality, and its absence is telling us that all of these original fermions have been gapped at the scale $\sim \Lambda$. Nonetheless, there are collective modes of these original fermions – those described by $\chi_8$ – which remain with much lower mass $\sim m$.

This last remaining fermion can now also be integrated out. However, crucially, a single Majorana fermion sits in one of two phases, depending on the sign of the mass. When $m > 0$, this fermion lies in the trivial phase. Integrating it out leaves us with a $\mathbf{Z}_2$ gauge theory:

$$B > 0: \qquad S_{\mathbf{Z}_2} = i\pi a \cup A.$$

This is the same $\mathbf{Z}_2$ gauge theory that we encountered previously in (5.21); once again we see that the $\mathbf{Z}_2^A$ global symmetry is spontaneously broken.

For $m < 0$, something different happens. Now the fermion lies in a topological phase [36]. This fact, which is reviewed in the Appendix, is reflected in the generation of an Arf invariant when the fermion is integrated out [10]. Now the low-energy $\mathbf{Z}_2$ gauge theory has the action

$$B < 0: \qquad S_{\mathbf{Z}_2} = i\pi\Big(a \cup A + \text{Arf}[a \cdot \rho_{NS}]\Big).$$

The extra Arf term for the dynamical gauge field changes the physics significantly. It has the effect of projecting out one of the holonomies, leaving us with a unique ground state. Indeed, in this case we can further integrate out the dynamical gauge field $a$, resulting in the effective topological term $S_{\text{eff}} = i\pi \text{Arf}[A \cdot \rho_{NS}]$. This is the promised symmetric mass generation: we are left with a gapped theory in which both $\mathbf{Z}_2^V$ and $\mathbf{Z}_2^A$ survive as global symmetries.

# Acknowledgements

We are grateful to Andreas Karch for collaboration on the earlier, related project [15], and to Philip Boyle-Smith, Michael Green, Masazumi Honda, and Shlomo Razamat for useful discussions. We are supported by the STFC consolidated grant ST/P000681/1. DT is a Wolfson Royal Society Research Merit Award holder and is supported by a Simons Investigator Award. CPT is supported by a Junior Research Fellowship at Gonville & Caius College, Cambridge.

# A    Appendix: The Arf Invariant

The Arf invariant is a mod 2 topological invariant which, on a Riemann surface with spin structure $\rho$, coincides with the mod 2 index of the chiral Dirac operator [35].

Specifically, the number of zero modes of the chiral Dirac operator $\slashed{D}_\rho$ is either even (typically none) in which case $\text{Arf}[\rho] = 0$, or odd (typically one) in which case $\text{Arf}[\rho] = 1$. For the purposes of this paper, we are interested in Majorana fermions on the torus. In this case, there are four spin different structures, each of which specifies whether the fermions have periodic (P) or anti-periodic (A) boundary conditions around each of the two cycles of the torus. The Arf invariant is

$$\text{Arf}[\rho] = \begin{cases} 1 & \rho = PP \\ 0 & \text{otherwise} \end{cases}.$$

The Arf invariant plays in important role in the theory of a Majorana fermion in $d = 1 + 1$ dimensions. Recall that a massive Majorana fermion in $d = 1 + 1$ dimensions has two phases, depending on the sign of the mass [36]. The simplest way to see this is to introduce a domain wall that interpolates from $+m$ to $-m$; the existence of the Jackiw-Rebbi zero mode is a signal that the two phases on either side differ. One of these phases is the "trivial" phase, and the other "topological".

In the absence of a domain wall (or a boundary) it is rather more subtle to distinguish these two different phases. Nonetheless, there is a way. Consider a Majorana fermion with mass $m$ on a Riemann surface $X$, endowed with a spin structure $\rho$. The partition function is given by

$$Z_{\text{Maj}}[\rho; m] = \text{Pf}\left(\slashed{D}_\rho + m\gamma^3\right).$$

The Pfaffian is naturally real, but there is no canonical choice of sign. If we define the partition function to be positive for $m > 0$, this is then the "trivial phase". The sign of the partition function for $m < 0$ then depends on the number of zero modes of that arise at $m = 0$. This is determined by the Arf invariant, and we have

$$Z_{\text{Maj}}[\rho; -m] = (-1)^{\text{Arf}[\rho]} Z_{\text{Maj}}[\rho; m].$$

Suppose that we define the partition function so that the theory is trivial when $m > 0$. Then if we integrate out the fermion with $m < 0$, we will be left with the Arf invariant $\text{Arf}[\rho]$ in the effective action, reflecting the fact that the fermion sits in the topological phase. In other words, the Arf invariant can be viewed as the partition function of a gapped, Majorana fermion in the topological phase [10]. The significance of these terms in various contexts have been discussed recently in a number of papers [11–15, 37–40].

The role of the Arf invariant becomes more prominent for fermions coupled to $\mathbf{Z}_2$ gauge fields. As we explained in the main text, in this case the spin structure $\rho$ is replaced by $a \cdot \rho$. If we have a single Majorana fermion of mass $m$, and subsequently integrate it out, we are left with an effective action for the $\mathbf{Z}_2$ gauge field, given by

$$S_{\text{eff}}[a] = \begin{cases} 0 & m > 0 \\ i\pi \, \text{Arf}[a \cdot \rho] & m < 0 \end{cases}.$$

This is reminiscent of the the way a Chern-Simons term is generated in $d = 2 + 1$ dimensions, depending on the sign of the mass of a fermion. Indeed, when it comes to duality, the Arf term plays a role analogous to the Chern-Simons term.

There are a number of properties of $\mathbf{Z}_2$ gauge fields and Arf invariants that we use throughout the paper. First,

$$\text{Arf}[(a+b)\cdot\rho] = \text{Arf}[a\cdot\rho] + \text{Arf}[b\cdot\rho] + \text{Arf}[\rho] + \int a\cup b. \tag{A.22}$$

Second, if we sum over a dynamical $\mathbf{Z}_2$ gauge field, the result depends crucially on whether there is an Arf term present. In the absence of an Arf term, the gauge field acts like a Lagrange multiplier, and on a Riemann surface of genus $g$ we have

$$\frac{1}{2^g}\sum_a (-1)^{a\cup V} = \begin{cases} 2^g & \text{if } V = 0 \\ 0 & \text{otherwise} \end{cases}.$$

However, if the Arf term is present, we instead get

$$\frac{1}{2^g}\sum_a (-1)^{a\cup V + \text{Arf}[a\cdot\rho] + \text{Arf}[\rho]} = (-1)^{\text{Arf}[V\cdot\rho]}. \tag{A.23}$$

This last fact will be important in Section 5, where the $\mathbf{Z}_2$ gauge fields play an important role in determining the phase of the theory.

Our focus on this paper is on the special things that happen with eight Majorana fermions. In particular, the work of Fidkowski and Kitaev showed that the $\mathbf{Z}$ classification of $d = 1+1$ topological insulators with time-reversal $T^2 = 1$ is broken by interaction to $\mathbf{Z}_8$. As explained in [10], this phase is diagnosed by a refinement of the Arf invariant, known as the Arf-Brown-Kervaire (ABK) invariant which is valued in $\mathbf{Z}_8$.

The ABK invariant arises as the partition function of a single Majorana fermion on an *unoriented* manifold with a Pin$^-$ structure [10]. In particular, on $\mathbf{RP}^2$ there are two Pin$^-$ structures, and the partition function is given by $e^{\pm i\pi/4}$.

# B  Appendix: More on Triality

In this appendix we include more details about various aspects of the triality.

## B.1  Triality of the Chiral Spectrum

We start by showing explicitly how the low-lying states match on either side of the chiral triality (3.14),

$$\begin{aligned}
\Theta_v[A\cdot\rho_{NS};q,z] &= \frac{1}{2}\sum_{a_0,a_1=0}^1 (-1)^{a\cup A + \text{Arf}[a\cdot\rho_{NS}]}\,\Theta_s[a\cdot\rho_{NS};q,z] \\
&= \frac{1}{2}\sum_{a_0,a_1=0}^1 (-1)^{a\cup A + \text{Arf}[A\cdot\rho_{NS}]}\,\Theta_c[a\cdot\rho_{NS};q,z].
\end{aligned}$$

We deal with anti-periodic and periodic boundary conditions in turn.

**Anti-Periodic Boundary Conditions**

First, set $A_1 = 0$, so the free fermions experience anti-periodic boundary conditions on the spatial circle. The expansion of the partition function is

$$\begin{aligned}
\Theta_v[A\cdot\rho_{NS};q,z] = q^{-1/6}\big[1 &+ (-1)^{A_0}\chi_{8_v}q^{1/2} + \chi_{28}q + (-1)^{A_0}(\chi_{8_v} + \chi_{56_v})q^{3/2} \\
&+ (1 + \chi_{28} + \chi_{35_v} + \chi_{35_s} + \chi_{35_c})q^2 + \dots\big],
\end{aligned}$$

where $\chi_{\mathbf{R}}$ is the character of the representation $\mathbf{R}$. For example, $\chi_{\mathbf{8_v}} = \sum_{\lambda \in \mathbf{8_v}} z^\lambda$. We now explain how this spectrum arises for the different theories in the triality.

All theories have a unique ground state. For the $\Theta_s$ and $\Theta_c$ theories, this arises in the $a_1 = 0$ sector where the fermions are anti-periodic.

The terms with integer powers of $q$ are simplest to describe since these arise from an even number of excitations above the anti-periodic sectors. At level $q$, we have just the $\mathbf{28}$ states. In the $\Theta_v$ theory, these arise from the anti-symmetrized part of the product $\mathbf{8_v} \otimes \mathbf{8_v} = \mathbf{1} \oplus \mathbf{28} \oplus \mathbf{35_v}$ while in both $\Theta_s$ and $\Theta_c$ theories these arise in the $a_0 = 0$ sector from the anti-symmetrized product of $\mathbf{8_s} \otimes \mathbf{8_s}$ and $\mathbf{8_c} \otimes \mathbf{8_c}$ respectively. Meanwhile, at level $q^2$, the $\Theta_v$ theory has $\mathbf{1} \oplus \mathbf{28} \oplus \mathbf{35_v}$ from two fermionic excitations with a derivative (or angular momentum excitation) and $\mathbf{35_s} \oplus \mathbf{35_c} \subset \mathbf{28} \otimes \mathbf{28}$ from the four fermion sector. There is a similar story for the $\Theta_s$ and $\Theta_c$ theories. Note that the expression multiplying integer powers of $q$ is always triality invariant.

The half-integer powers of $q$ are more involved. These arise from an odd number of fermionic excitations in the $\Theta_v$ theory. In the $\Theta_s$ and $\Theta_c$ theories, where $(-1)^F$ is gauged, states with an odd number of fermionic excitations in the anti-periodic $a_1 = 0$ sector are always projected out. However, the situation is different in the $a_1 = 1$ sector, where the fermions are periodic.

At level $q^{1/2}$, the $\Theta_v$ theory clearly has a single $\mathbf{8_v}$ state with $(-1)^F = (-1)^{A_0} = -1$. In the $\Theta_s$ theory, this state arises from the periodic sector with $a_1 = 1$. Here the ground states lie in $\mathbf{8_c} \oplus \mathbf{8_v}$; the $\mathbf{8_c}$ has $(-1)^F = +1$ and the $\mathbf{8_v}$ has $(-1)^F = -1$. The presence of the $\mathrm{Arf}[a \cdot \rho_{NS}]$ term means that we project onto those states with $(-1)^F = -1$; these are precisely the $\mathbf{8_v}$ ground states. There is a similar story in the $\Theta_c$ theory. Now the ground states are in $\mathbf{8_v} \oplus \mathbf{8_s}$ but the lack of $\mathrm{Arf}[a \cdot \rho_{NS}]$ term means that this time we project onto states with $(-1)^F = +1$. This is again leaves us with a $\mathbf{8_v}$ ground state.

At level $q^{3/2}$, the $\mathbf{8_v} \oplus \mathbf{56_v}$ come from the anti-symmetrized product of $\mathbf{8_v} \otimes \mathbf{8_v} \otimes \mathbf{8_v}$ in the $\Theta_v$ theory. Meanwhile, in the $\Theta_s$ theory, we can excite a single fermion above the $\mathbf{8_c}$ ground state. This too gives $\mathbf{8_c} \otimes \mathbf{8_s} = \mathbf{8_v} \oplus \mathbf{56_v}$. Similarly, in the $\Theta_c$ theory, we can excite a single fermion above $\mathbf{8_s}$ ground state, again giving $\mathbf{8_s} \otimes \mathbf{8_c}$.

**Periodic Boundary Conditions**

Now set $A_1 = 1$, ensuring that the free fermions have periodic boundary conditions around the spatial circle. This time the $\Theta_v$ partition function is

$$\Theta_v[A \cdot \rho_{NS}; q, z] = q^{1/3} \left[ (\chi_{\mathbf{8_s}} + (-1)^{A_0} \chi_{\mathbf{8_c}}) + \left[ (-1)^{A_0} (\chi_{\mathbf{8_c}} + \chi_{\mathbf{56_c}}) + (\chi_{\mathbf{8_s}} + \chi_{\mathbf{56_s}}) \right] q + \dots \right],$$

and the ground states sit in $\mathbf{8_s} \oplus \mathbf{8_c}$. We should first understand how this arises in the $\Theta_s$ and $\Theta_c$ theories. In both, the $a \cup A$ term now provides an extra $(-1)^F$ charge to states. This means that in the $a_1 = 0$ sector the ground state is projected out and the surviving states have $(-1)^F = -1$, meaning that we must excite an odd number of fermions. This provides half of the ground states in the above partition function: the $\mathbf{8_s}$ in the $\Theta_s$ theory, and the $\mathbf{8_c}$ in the $\Theta_c$ theory. The other half of the ground states come from the $a_1 = 1$ sector, where the fermions have periodic boundary conditions. The net result of the $a \cup A$ and $\mathrm{Arf}[a \cdot \rho_{NS}]$ terms is to keep precisely the states that we need.

At level $q$, we excite a single fermion in the $\Theta_v$ theory above one of the two sets of ground states, giving $\mathbf{8_s} \otimes \mathbf{8_v} = \mathbf{8_c} \oplus \mathbf{56_c}$ and $\mathbf{8_c} \otimes \mathbf{8_v} = \mathbf{8_s} \oplus \mathbf{56_s}$. In the $\Theta_s$ theory, when $a_1 = 1$, the excitation of a single fermion above the $\mathbf{8_v}$ ground state gives us the $\mathbf{8_v} \otimes \mathbf{8_s}$ states. Meanwhile, in the $a_0 = 0$ sector we get states at this level from the anti-symmetrized product of $\mathbf{8_s} \otimes \mathbf{8_s} \otimes \mathbf{8_s}$, which gives the remaining $\mathbf{8_s} \oplus \mathbf{56_s}$. There is a similar story in the $\Theta_c$ theory.

## B.2 Comments on the $SO(8)/\mathbf{Z}_2$ Triality

The self-triality of the $SO(8)/\mathbf{Z}_2$ was written in (3.18) as

$$\mathcal{Z}_{v/\mathbf{Z}_2}[V,A;\rho_{NS}] = \mathcal{Z}_{s/\mathbf{Z}_2}[V+A,V;\rho_{NS}] = \mathcal{Z}_{c/\mathbf{Z}_2}[A,V+A;\rho_{NS}], \tag{B.24}$$

with $V$ and $A$ background fields coupling to the global $\mathbf{Z}_2$ symmetries in each theory. Here we describe a number of additional properties of this triality.

**More Symmetries**

Our theory has two further $\mathbf{Z}_2$ symmetries, both of which exhibit interesting 't Hooft anomalies. First, the equivalence (B.24) only exhibits the cyclic permutations of $(\mathbf{8_v}, \mathbf{8_s}, \mathbf{8_c})$. This corresponds to the subgroup $\mathbf{Z}_3 \subset S_3$ of the full triality group, which is $S_3 = \mathbf{Z}_3 \rtimes \mathbf{Z}_2$. We can ask: how does the remaining $\mathbf{Z}_2$ act? For example, we can ask how the transformation $\mathbf{Z}_2^{\text{flip}} : \mathbf{8_s} \longleftrightarrow \mathbf{8_c}$ (which, in terms of fugacities, is $z_3 \longleftrightarrow z_4$) acts on the $\mathbf{8_v}$ partition function.

This, it turns out, is not quite trivial since it acts on the ground states in the periodic sector. From (3.9) and (3.11), the effect of this transformation is simply to change the sign of the totally periodic partition function,

$$\mathbf{Z}_2^{\text{flip}} : \Theta_v[\rho] \;\mapsto\; (-1)^{\text{Arf}[\rho]} \Theta_v[\rho].$$

Acting on the $SO(8)/\mathbf{Z}_2$ partition function, this becomes

$$\mathbf{Z}_2^{\text{flip}} : \mathcal{Z}_{v/\mathbf{Z}_2}[V,A;\rho_{NS}] \;\mapsto\; (-1)^{A\cup V} \mathcal{Z}_{v/\mathbf{Z}_2}[V+A;A;\rho_{NS}].$$

The existence of the $A \cup V$ term can be viewed as a mixed 't Hooft anomaly between $\mathbf{Z}_2^{\text{flip}}$ and $\mathbf{Z}_2^V \times \mathbf{Z}_2^A$. Note that the $S_3$ triality group acts not only on the Spin(8) representations, but also on the pair of $\mathbf{Z}_2$ gauge fields, $A$ and $V$. This reflects the isomorphism $S_3 \cong SL(2;\mathbf{Z}_2)$, which has a natural action on $(V,A)$, viewed as a pair of $\mathbf{Z}_2$-valued objects.

Second, we can look at time reversal. This is an anti-unitary symmetry which acts on the partition function by complex conjugation. The partition functions $\mathcal{Z}_v$ and $\mathcal{Z}_{v/\mathbf{Z}_2}$ are almost real; in the presence of background gauge fields, they change only by an overall sign,

$$T : \mathcal{Z}_{v/\mathbf{Z}_2}[V,A;\rho] \;\mapsto\; (-1)^{A\cup V} \mathcal{Z}_{v/\mathbf{Z}_2}[V,A;\rho].$$

Once again, there is a mixed 't Hooft anomaly with $\mathbf{Z}_2^V \times \mathbf{Z}_2^A$. However, the same factor of $(-1)^{A\cup V}$ arises in both $\mathbf{Z}_2^{\text{flip}}$ and $T$ anomalies, telling us that the combination of the two is anomaly-free.

**Triality on General Spin Structures**

The triality (B.24) makes reference to a specific spin structure $\rho_{NS}$ on the torus. This can be traced to number of sign choices that were made when the left- and right-handed partition functions were combined. The presence of this preferred spin structure means that it is not obvious how this triality extends to more general Riemann surfaces. It seems reasonable that on a general Riemann surface, a preferred reference spin structure (and a preferred choice of cycles in the first homology, a so-called *marking*) must be chosen to formulate such a duality. The reason is that such a choice is required in order to unambiguously define the phase of the chiral partition function.

On the torus, we write a general spin structure as $\rho = R \cdot \rho_{NS}$ and then define the general partition function

$$\mathcal{Z}_{v/\mathbf{Z}_2}[V,A;\rho] = \frac{1}{2}\sum_{a}(-1)^{R\cup A + a\cup V + \text{Arf}[(V+A)\cdot\rho] + \text{Arf}[\rho]} \mathcal{Z}_v[a,A;\rho]$$

and similar for the $\mathbf{8_s}$ and $\mathbf{8_c}$ partition functions. The presence of the $R \cup A$ means that the triality (B.24) generalizes to

$$\mathcal{Z}_{v/\mathbf{Z}_2}[V,A;\rho] = \mathcal{Z}_{s/\mathbf{Z}_2}[V+A,V;\rho] = \mathcal{Z}_{c/\mathbf{Z}_2}[A,V+A;\rho].$$

The $R \cup A$ term has fixed up ambiguities in the signs in the partition function, ambiguities that were absent when $A = 0$. A similarly consistent choice of signs is needed to define the triality on a general Riemann surface.

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
