# Peer review of "Notes on 8 Majorana Fermions"

_SciPost Physics Lecture Notes, doi:SciPost Phys. Lect. Notes 14 (2020)_

## Round 1 · Referee Report · Anonymous · 2020-2-6

Strengths
A very clear and pedagogical discussion of the intricacies of bosonization in 1+1d and SO(8) triality, with applications to Symmetry Protected Topological Phases.
Report
The paper is a helpful guide to understanding the physics of bosonization in 1+1d using eight massless Majorana fermions as a particularly interesting example. Much of the paper is an exposition of well-known results used heavily in perturbative string theory. But the paper is still a valuable contribution, since it explains various subtle points using a more modern language (such as the Arf invariant). Also, the authors emphasize that there are several versions of triality only some of which appear in string theory applications. The connection with SPT phases in 2+1d is outlined in the last section.
Requested changes
Typo on the bottom of page 2: 8_v should be 8_c.

---

## Round 1 · Referee Report · Anonymous · 2020-2-23

Strengths
1- Uniform presentation of some results in high energy and condensed matter literature.
2- Pedagogical exposition.
Weaknesses
Nothing in particular.
Report
In this paper, the authors discuss various subtleties associated with $SO(8)$ triality. To begin, they clarify that triality is really only a property of $Spin(8)$ and $SO(8)/ \mathbb{Z}_2$. Though one might have naively expected triality to imply the equivalence of theories of free fermions in the $\mathbf{8}_v$, $\mathbf{8}_s$, and $\mathbf{8}_c$ representations, they show that this is not always the case. Instead, equivalence between theories is obtained only when fermions are coupled to appropriate $\mathbb{Z}_2$ gauge fields.
They then discuss two applications of triality. In the string theory context, $Spin(8)$ triality is crucial for showing the equivalence of the Green-Schwarz and RNS formulations of Type $\mathrm{II}$ strings. The $SO(8)/ \mathbb{Z}_2$ triality plays an analogous role for Type $0$ strings. In the condensed matter context, triality is useful in understanding how systems of eight fermions can become gapped without breaking chiral symmetries, connecting to well-known results of Fidkowski and Kitaev.
Though much of the material presented here has appeared elsewhere, this paper serves as a pedagogical exposition of the topic, and puts various results in a more modern language (i.e. the language of discrete gauge theory and topological phases). As such, I recommend this paper for publication.
Requested changes
Before publication, I would like to suggest the following very minor changes:
1- At the bottom of page 2, $\mathbf{8}_v$ should be $\mathbf{8}_c$.
2- The definition of $\eta$ on page 8 should be $\eta = q^{1/24} \prod_{n=1}^\infty (1-q^n)$.
3- In the formula for $Z_{IIA}$ on page 14, there seems to be a minus sign missing.
4- In the formula after (A.1), it would be helpful to define $g$ as the genus of the Riemann surface.
5- In the last sentence of Appendix A, it is claimed that the value of the $\mathrm{ABK}$ invariant on $\mathbb{RP}^2$ is $e^{i \pi /4}$. This is not completely correct. $\mathbb{RP}^2$ admits two $\mathrm{Pin}^-$ structures, and for only one of them is this the correct result. (For the other $\mathrm{Pin}^-$ structure, the correct result is $e^{-i \pi /4}$)

---

## Round 2 · Author Response

Contains the small changes suggested by the referees.

You are currently on this page

Resubmission 1906.07199v2 on 5 March 2020

---

## Editorial Decision

published